# Understanding Adversarial Transferability in Federated Learning

## Abstract

With the promises Federated Learning (FL) delivers, various topics regarding its robustness and security issues have been widely studied in recent years: such as the possibility to conduct adversarial attacks (or transferable adversarial attacks) in a while-box setting with full knowledge of the model (or the entire data), or the possibility to conduct poisoning/backdoor attacks during the training process as a malicious client. In this paper, we investigate the robustness and security issues from a different, simpler, but practical setting: a group of malicious clients has impacted the model during training by disguising their identities and acting as benign clients, and only revealing their adversary position after the training to conduct transferable adversarial attacks with their data, which is usually a subset of the data that FL system is trained with. Our aim is to offer a full understanding of the challenges the FL system faces in this setting across a spectrum of configurations. We notice that such an attack is possible, but the federated model is more robust compared with its centralized counterpart when the accuracy on clean images is comparable. Through our study, we hypothesized the robustness is from two factors: the decentralized training on distributed data and the averaging operation. Our work has implications for understanding the robustness of federated learning systems and poses a practical question for federated learning applications.

## 1 Introduction

The ever-growing usage of mobile devices such as smartphones and tablets leads to an explosive amount of distributed data collected from user-end. Such private and sensitive data, if can be fully utilized, will greatly improves the power of more intelligent systems. Federated learning (FL) provides a solution for decentralized learning by training quality models through local updates and parameter aggregation McMahan et al. (2017). A FL system maintains a loose federation of participated clients and a centralized server that holds no data but the aggregated model. During training, the central server distributes the global model to a random subset of participants where they updates the model with the private data locally and submits the updated model back to the server for aggregation (*e.g.* average) at each round. By design, the system has no visibility to the local data, allowing it to benefit from a wide range of private data while maintaining participant privacy, and the averaging provides an efficient way to leverage the updated parameters compared with distributed SGD.

Despite the fact that FL protects privacy, the loose organization and its invisibility to local data makes it more vulnerable to the various attacks including data poisoning Huang et al. (2011), model poisoning (backdoor attack) Bhagoji et al. (2019); Bagdasaryan et al. (2020), free-riders attack Lin et al. (2019) and various reconstruction attack that leaks the data and privacy of individual participants Geiping et al. (2020); Zhu et al. (2019). Various anomaly detection based methods have been proposed to prevent possible poisoning attacks such as Byzantine-tolerant aggregation Yin et al. (2018), clustering-based selection Shen et al. (2016) and anomaly detection in spectral domain Li et al. (2020). Reputation is introduced to prevent free-rider Xu & Lyu (2020) and differential privacy techniques are leveraged to preserve the privacy against GANs-based reconstruction attacks Augenstein et al. (2019); Hao et al. (2019). Another line of FL security research focuses on the attack during inference, i.e., the adversarial attack Biggio et al. (2013); Szegedy et al. (2013). Same as any other deep learning application, federated systems are also found vulnerable to the adversarial examples carefully crafted to deceive the model Zizzo et al. (2020). FAT first discusses the possibility

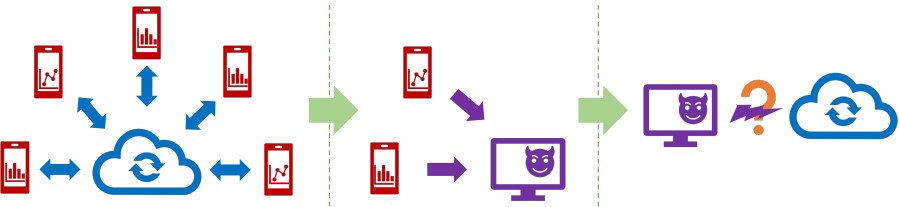

Figure 1: The practical setting of our interest: some clients are disguised as normal clients and participate the training process of a FL system regularly, but later contribute the data to train a malicious model for attacking the trained FL system. This paper studies the possibility of such attack across different configurations of the FL system.

to attack the federated model with adversarial examples and the interplay of adversarial training with FL Zizzo et al. (2020). What is following is a line of research proposing more robust FL method tailored to defend the adversarial examples Zhou et al. (2020); Reisizadeh et al. (2020); Hong et al. (2021). Despite great progresses made, the community mainly focuses on a white-box setting where the attacker gets full access to the various aspects of the target model such as the gradient and the output of the model. However, in most real-world scenarios where FL system, *e.g.* Gboard Hard et al. (2018), is trained and deployed, adversary have no access to any knowledge of the model or the full training set.

Different from the above two settings, we notice that FL is naturally raising another security and robustness challenge: during the training, the malicious client can disguise to be a benign one to contribute regularly to the update of the model parameters, and only reveal its adversary identity after the training. Thus, the client naturally obtains a subset of the data used to train the FL model, and has the potential to exploit this slice of data for adversarial attacks. While most of the federated application poses no criteria for eliminating the hostile participants Hard et al. (2018), even if there are selection mechanisms (*e.g.* Krum for defending backdoor attack) Fang et al. (2020); Li et al. (2020); Bagdasaryan et al. (2020), the attacker undoubtedly escapes from it since no hostile actions are performed during training. The attacker, after acquiring the data, can train a substitute model to perform the transfer-based black-box attack.

In this paper, we take the first step to explore this practical perspective of robustness in FL. Stemming from the above discussed scenarios, we propose a simple yet practical assumption: the attacker possesses some but limited amount of the users' data but no knowledge about the target model or the full training set. To better understand and evaluate the robustness of current FL systems and provide implications for future research to improve the security on this regard, we conduct investigations on the adversarial transferability under FL settings. First, we establish baseline models with ResNet50 on CIFAR10 to provide preliminary understandings about the robustness of FL under white-box attack. Then we evaluate the transferability of adversarial examples generated from different source models to attack a federated-trained model. We further investigate two properties of FL, namely the decentralized training and the averaging operation, and their correlation with federated robustness. We have the following findings:

- We find that, while there is indeed security challenges of this setting (*i.e.* the malicious clients can attack the federated model after the training through transferable adversarial examples), the federated model is more robust under white-box attack compared with its centralized-trained counterpart when their accuracy on clean images are comparable.
- We investigate the transferability of adversarial examples generated from models trained by various number of users' data. We observe that, without any elaborated techniques such as dataset synthesis Papernot et al. (2017) or attention Wu et al. (2020), a regularly trained source model with only limited users' data can perform transfer attack. With ResNet50 on CIFAR10 datatset, we achieve an transfer rate of over 60% with only 10% of the total clients and an transfer rate of almost 90% with 20%. With strong augmentation, source model can attack with a transfer rate of almost 70% and 80% using only 5% and 7% of the total users.
- We investigate two intrinsic properties of the FL, namely the property of distributed training and the averaging operation and discover that both heterogeneity and dispersion degree of the decentralized data as well as the averaging operations can significantly decreases the transfer rate of transfer-based black-box attack.

## 2 BACKGROUND

### 2.1 ADVERSARIAL ROBUSTNESS

The adversarial robustness of a model is usually defined as the model's ability to predict consistently in the presence of small changes in the input. Intuitively, if the changes to the image are so tiny that they are imperceptible to humans, these perturbation will not alter the prediction of the model. Formally, given a well trained classifier $f$ and image-label pairs $(x, y)$ on which the model correctly classifies $f(x) = y$, $f$ is defined to be $\epsilon$-robust with respect to distance metric $d(;)$ if

$$\mathbb{E}_{(x,y)} \min_{x':d(x',x)\leq\epsilon} \alpha(f(x';\theta), y) = \mathbb{E}_{(x,y)} \alpha(f(x;\theta), y) \tag{1}$$

which is usually optimized through maximizing the following:

$$\mathbb{E}_{(x,y)} \min_{x':d(x',x)\leq\epsilon} \alpha(f(x';\theta), y) \tag{2}$$

where $\alpha$ denotes the function evaluating prediction accuracy. In the case of classification, $\alpha(f(x;\theta), y)$ yields 1 if the prediction $f(x;\theta)$ equals ground-truth label $y$, 0 otherwise. The distance metric $d(;)$ is usually approximated by $L_0$, $L_2$ or $L_\infty$ to measure the visual dissimilarity of original image $x$ and the perturbed image $x'$. Despite the change to the input is small, the community have found a class of methods that can easily manipulate model's predictions by introducing visually imperceptible perturbations in images Szegedy et al. (2013); Goodfellow et al. (2014); Moosavi-Dezfooli et al. (2016). From the optimization standpoint, it is achieved by maximizing the loss of the model on the input (Madry et al., 2017):

$$\max_\delta l(f(x+\delta;\theta), y) \qquad s.t. \ d(x+\delta, x) < \epsilon \tag{3}$$

where $l(\cdot, \cdot)$ denotes the loss function (*e.g.* cross entropy loss) for training the model $f$ parameterized by $\theta$. While these attack methods are powerful, they usually require some degrees of knowledge about the target model $f$ (*e.g.* the gradient). Arguably, for many real-world settings, such knowledge is not available, and we can only expect less availability of such knowledge on FL applications trained and deployed by service providers. On the other hand, the hostile attacker having access to some but limited amount of users' data is a much more realistic scenario. Thus, we propose the following assumption for practical attack in FL: given the data of $n$ malicious users $D_m = \bigcup_{i=1}^n D_i$ where $D_i = \{(x_k, y_k)|k = 1, \cdots, m_i\}^{(i)}$ contains $m_i$ data points, we aim to acquire a transferable perturbation $\delta$ by maximizing the same objective as in Equation 3 but with a substitute model $f'$ trained by $D_m$:

$$\delta = \arg\min_\delta l(f'(x+\delta;\theta), y) \qquad s.t. \ d(x+\delta, x) < \epsilon \tag{4}$$

We hope to test whether this $\delta$ can be used to deceive the target model $f$ as well.

### 2.2 THE SECURITY AND ROBUSTNESS OF FEDERATED LEARNING

FL provides one of the most promising solutions to take advantage of the large amount of distributed data collected from user-end without violating privacy requirements. However, the loose organization and its invisibility to local data makes it more vulnerable to the various attacks.

**Poisoning and Backdoor Attack.** Poisoning attack including data Biggio et al. (2012) and model poisoning attack Fang et al. (2020) aims to maximize the classification error of global model via malicious data injection or local model parameter manipulation on compromised client devices. Backdoor attack, instead, aims to inject a malicious task into the existing model while retaining the accuracy of the actual task Bagdasaryan et al. (2020); Sun et al. (2019). Chen et al. (2017) propose the model replacement to replace the global model with a backdoored model stealthily. Fung et al. (2018) introduce label-flipping to backdoor the federated model while Xie et al. (2019) uses GANs to attack the global model. Boosting weights and including stealth into the adversarial objective also proves to be effective to backdoor Bhagoji et al. (2019). To defend against these attacks, various anomaly detection based methods have been proposed. One of the earliest is the Byzantine-tolerant (or Byzantine-resistant) aggregation Shejwalkar & Houmansadr (2021) such as Krum, MultiKrum Blanchard et al. (2017), Bulyan Guerraoui et al. (2018), Trimmed-mean and Median Yin et al. (2018), which focuses mainly on geometric distance between the hostile gradients and benign ones.

Shen et al. (2016) uses clustering to distinguish the malicious updates from a collaborate of clients while Fang et al. (2020) proposes Error Rate and Loss based Rejection which rejects updates that has large influence to either error rate or model loss on the validation set. Li et al. (2020); Kieu et al. (2019) learns another model to detect malicious updates in the spectral domain. Zawad et al. (2021) discovers that the data heterogeneity of FL largely impacts the robustness and attack performance of backdoor attack. Knowledge distilliation and prunning is also proven effective to prevent these attacks Li & Wang (2019); Liu et al. (2018).

**Adversarial and Transfer Attack.** Adversarial attack refers to a type approach that generates imperceptible perturbations which leads to mis-classification with high confidence Szegedy et al. (2013); Biggio et al. (2013). The community has discovered various types of powerful attack methods such as FGSM Goodfellow et al. (2014), PGD Madry et al. (2017), deepfool Moosavi-Dezfooli et al. (2016) and many others Carlini et al. (2019); Xiao et al. (2018); Dong et al. (2018). Despite that machine learning models are highly susceptible to adversarial examples, these attacks require various knowledge of the model to ensure a successful attack, which is impractical in real-world scenarios. Consequently, black-box attack is introduced which can be divided into query-based and transfer-based attack. Query-based attack requires queries to the target model and treat the prediction as the ground-truth label forming the initial training set to train the substitute model Papernot et al. (2017). Augmentation and regularization are leveraged to further improve the attack Guo et al. (2019); Wang et al. (2021). Despite the queries can help the source model to learn a decision boundary similar to one of the target model, the requirement for query make it impractical to attack systems with access limit. Transfer-based, on the other hand, leverage the full training set used by the target model to train a substitute model Zhou et al. (2018). The condition for transfer attack in practice is also difficult to achieve since it's even more difficult or costly and laborsome to access the original training set or collecting a similar new one, especially in FL where data privacy are carefully protected. Another research perspective is the mechanism behind the black-box attack. Black-box attack builds upon an intriguing discovery that adversarial examples are higher transferable even between models of different architectures Szegedy et al. (2013); Goodfellow et al. (2014). This transferability can be partly explained by the similarity between the source and target models. Goodfellow et al. (2014); Liu et al. (2016) shows that adversarial perturbations are highly aligned with the weight vectors of a model and that different models learn similar decision boundaries when trained to perform the same task. Tramèr et al. (2017) discovers that adversarial examples span a contiguous subspace of large dimensionality and the shared subspace of two models enables the transferability. While Ilyas et al. (2019) shows that adversarial perturbations are non-robust feature captured by the model, Waseda et al. (2021) uses this theory to explain why different mistakes exists in the transfer attack. Demontis et al. (2019) demonstrates that similar gradient of source and target model and a low variance of loss landscapes leads to higher possibility to transfer attack.

**Discussion.** In this paper, we adopt the idea of transfer-based attack into the practical setting of FL by proposing a simple yet realistic attack setting. Here, we discuss the key differences and the significance of our setting compared with others.

**Key Difference 1:** Different from query-based or transfer-based black-box attack, we assume the malicious clients possess the data themselves, impacting the target model during training and attack during inference. We also present a comparison of our attack setting and the query-based attack in Appendix A. Note that our attack setting doesn't contradict with the query-based attack. In fact, we can perform with both if the FL system allows a certain number of queries, which we leave to future works to explore.

**Key Difference 2:** Poisoning attack or backdoor attack manipulates the parameters update during target model training which can be defended by anomaly detection. Moreover, in practical, despite clients preserve the training data locally, the training procedure and communication with the server are highly encapsulated and encrypted with secret keys, which is even more unrealistic and laborsome to manipulate. Our attack setting circumvents this risk since no hostile action is performed during the training but successfully boost the attack possibility during inference time.

## 3 INVESTIGATION SETUP AND RESEARCH GOALS

With the possibility that malicious client disguise to be a benign during training and exploit the obtained data for adversarial attacks, as discussed in Section 1, it is of necessity to investigate the

robustness of federated model against transfer-based attack, *i.e.* transferability of adversarial examples against FL. More importantly, we want to understand where do the robustness of FL comes from and how the core components of FL effects its own robustness. To sum up, we put forward the following research goals:

**GOAL 1:** Investigate the possibility of a transfer attack with limited data and validate whether it is possible and practical for the attacker to lay benign during the training process and leverage the obtained data and performs the adversarial attack.

**GOAL 2:** Explore the adversarial examples generated by various models (trained in different paradigms) and their tranferability against the federated model.

**GOAL 3:** We aim to explore how different degree of decentralization and the heterogeneity of data affects the transfer attack. We also aim to investigate how the aggregation, *i.e.* average, influences the transferability of the adversarial examples against the FL model.

## 3.1 EXPERIMENT SETUP

**Threat Model.** We use PGD Madry et al. (2017) and follow Zizzo et al. (2020) to perform the PGD with 10 iterations, no random restart, and an epsilon of 8 / 255 over $L_\infty$ norm on CIFAR10.

**Settings.** We first build up the basic FL setting. We split the datatset into 100 partitions of equal size in an iid fashion. We adopt two models for the experiments: CNN from McMahan et al. (2017) since it is commonly used in the FL literature and the widely used ResNet50 He et al. (2016) which represents a more realistic evaluation. We conduct training in three paradigms: the centralized model, federated model and the source model with limited number of clients' data. For federated model, we use SGD without momentum, weight decay of 1e-3, learning rate of 0.1, local batch size of 50 following Acar et al. (2021). We train locally 5 epochs on ResNet50 and 1 epoch on CNN. For centralized and source model training, we leverage SDG with momentum of 0.9, weight decay of 1e-3, learning rate of 0.01 and batch size of 64. For adversarial training, we use the same setting as centralized and leverage PGD to perform the adversarial training. We refer to Zizzo et al. (2020) for the detail of adversarial training.

**Metrics.** Accuracy (Acc) and adversarial accuracy (Adv.Acc) is reported to reflect the performance and the robustness white-box attack. For adversarial transferability, we report transfer accuracy (T.Acc) and transfer success rate (T.Rate) as detailed in 3.2.

## 3.2 ADVERSARIAL TRANSFERABILITY IN FEDERATED LEARNING

To define the transferability of adversarial examples, we first introduce the definition of the source model, target model and the adversarial example. Source model is the substitute model used to generate adversarial examples while target model is the target aimed to attack. Given the validation set $x = \{(x_i, y_i)\}$, source model $f'$, target model $f$ and adversarial perturbation function $adv(\cdot, \cdot)$ (*e.g.* PGD), we first define the following sets: $s1 = \{x_i | f'(x_i) = y_i\}$, $s2 = \{x_i | f'(adv(x_i, f')) \neq y_i\}$, $s3 = \{x_i | f(x_i) = y_i\}$, $s4 = \{x_i | f(adv(x_i, f')) \neq y_i\}$. Adversarial examples are defined to be those samples that are originally correctly classified by model $f'$ but are mis-classified when adversarial perturbation is added, i.e., $s1 \cap s2$. Adversarial transferability against the target model refers to the ability of an adversarial examples generated from the source model to attack the target model (becomes an adversarial example of target model). We define transfer rate (T.Rate) to measure the adversarial transferability:

$$T.Rate = \frac{||s1 \cap s2 \cap s3 \cap s4||}{||s1 \cap s2 \cap s3||}$$

where $|| \cdot ||$ denotes the cardinality of one set. We also define transfer accuracy (T.Acc) as:

$$T.Acc = 1 - \frac{||s4||}{||x||}$$

## 3.3 ROBUSTNESS WITH COMPARABLE ACCURACY

We first provide preliminary understandings about the robustness of FL and train centralized model for 200 epochs and federated model for 400 rounds resulting in a decent accuracy of over 90%

(see the regular column of Table 1). For CNN model, we train 200 epochs for centralized model and 600 rounds for federated model to achieve and accuracy of over 75%. We can observe that both clean and adversarial accuracy of federated model is lower than its centralized counterpart, which is also reported in Zizzo et al. (2020). However, we conjecture that such increase in adversarial accuracy is not attributed to intrinsic robustness of centralized model but largely due to its high clean accuracy. To validate this hypothesis and facilitate a fair comparison between the two paradigms, we early-stop both models when their clean accuracy reach 90% (75% for CNN) and report the results in the same-acc column of Table 1. We early stop at 80% for adversarial training (72% for CNN). We can see, when both models reach a comparable clean accuracy, federated model shows greater robustness against white-box attack compared with the centralized model.

Table 1: Centralized and federated model under white-box attack.

| Paradigm | Architecture | same-acc | | regular | |
|---|---|---|---|---|---|
| | | Acc | Adv.Acc | Acc | Adv.Acc |
| centralized | R50 | 90.20 | 0.01 | 95.24 | 0.40 |
| | R50 (adv) | 81.23 | 23.27 | 89.46 | 46.09 |
| | CNN | 75.06 | 1.24 | 82.41 | 0.35 |
| | CNN (adv) | 73.15 | 20.89 | 76.78 | 28.92 |
| federated | R50 | 90.29 | 0.05 | 92.31 | 0.02 |
| | R50 (adv) | 80.05 | 36.44 | 81.05 | 39.11 |
| | CNN | 75.09 | 3.68 | 76.83 | 3.98 |
| | CNN (adv) | 72.85 | 25.5 | 72.87 | 24.35 |

Table 2: T.Rate and transfer accuracy of PGD attack between pairs of models using various training paradigms. The row and column denote the source and target model respectively. For each cell, the left is the transfer accuracy and the right is the T.Rate.

| | | federated | centralized |
|---|---|---|---|
| R50 | federated | 0.15 / 99.83 | 2.01 / 97.67 |
| | centralized | 24.28 / 71.94 | 7.41 / 91.48 |
| CNN | federated | 19.31 / 76.32 | 21.59 / 71.84 |
| | centralized | 30.57 / 56.59 | 22.62 / 68.19 |

## 4 EXPERIMENTS

After having some preliminary understandings regarding the robustness of FL under white-box attack, we discuss the transfer attack with limited data in this section. As described in Section 1, a much more realistic setting is that a group of hostile clients lay benign during training and leverage the fraction of training data to perform transfer attack. Consequently, the robustness against transfer attack of the FL model becomes vital in evaluating the security and robustness of FL applications. In this section, we first discuss the transfer attack in Section 4.1 and then explore the possibility of attacking with limited data in Section 4.2.

### 4.1 ROBUSTNESS AGAINST TRANSFER ATTACK

We explore the examples generated by two different training paradigms and their transferability when applying them to different models. Since similarity of decision boundary and clean accuracy influences and reflects the transferability between models Goodfellow et al. (2014); Liu et al. (2016); Demontis et al. (2019), we early stop both federated and centralized model at 90% of accuracy which means we terminate the training once its clean accuracy reaches 90% (75% for CNN). Note that we follow this training setting for the rest of this paper. From Table 2, we can see that the adversarial examples generated by federated model are **highly transferable** to both federated and centralized model while adversarial examples generated by centralized model exhibit less transferability. The T.Rate of federated-to-centralized attack is even larger than centralized-to-centralized attack. Secondly, T.Rate of adversarial examples between models trained under same paradigms is larger than models trained under different paradigms, which can be attributed to the difference of the two training paradigms, *e.g.* the discrepancy in the decision boundary Goodfellow et al. (2014); Liu et al. (2016) or different sub-space Tramèr et al. (2017).

### 4.2 TRANSFER ATTACK WITH LIMITED DATA

In this section, we will evaluate the possibility of our proposed attack setting, *i.e.* transfer attack with limited data. To simulate this scenario, we fixed the generated partition used in the federated training and randomly selects a specified number of users as malicious clients whose data is available for performing the attack. To perform the transfer attack, we trains a substitute model in an centralized manner with the collected data. Training details are specified in Section 3.1. One of the key differences of proposed setting and conventional transfer-based attack is number of available data to train the substitute (source) model. The number of clients used to train the source model

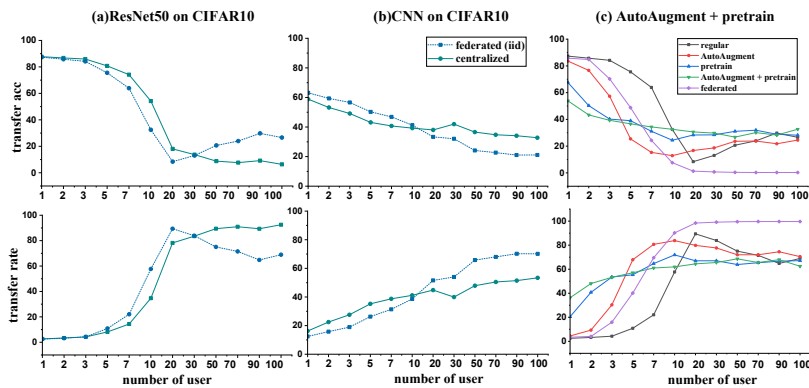

Figure 2: Attack with data from limited number of users.

also serves as a key factor for a successful attack since one would reason that more data will lead to higher success rate. To provide an overview of the transferability of adversarial examples generated by the source model trained with different number of clients, we plot the their relation in Figure 2. We have the following observations:

- **Observation 1.** T.Rate increases as the number of user used in source model increases. This is consistently observed in both centralized and federated model.
- **Observation 2.** With only 20% of clients the source model achieves an T.Rate of 90% and 50% with ResNet50 and CNN respectively. We notice that with ResNet50, the T.Rate of 20% clients is even larger than a transfer attack with full training data (71.94% T.Rate). With CNN, source model trained with 20% clients can achieve 50% T.Rate which only lags behind the transfer attack by 6% (56.59%).
- **Observation 3.** The T.Rate of adversarial examples against the centralized model keeps increasing while the T.Rate against the federated model increases till 20 user and decreases.

From observation 2, we can see that the proposed attack can achieve comparable or even better T.Rate or lower T.Acc. Consequently, we can conclude that the proposed attack setting with limited data is likely to cause significant security breach in the current and future FL applications. To further explain observations 1 and 3 that the T.Rate of ResNet50 model rises to the peak and then decreases, we hypothesize the following: when the number of clients used to train the source model is small, the clean accuracy of the source model is also low, leading to a large discrepancy in the decision boundary. Increasing the number of users used in source model minimizes such discrepancy until there are sufficient amount of data to train a source model with similar accuracy. At this point, the difference between the federated and centralized Caldarola et al. (2022) becomes dominant factor affecting the transferability since the source model is trained in centralized paradigm.

**Boosting transfer success rate with standardize training techniques.** Following this conjecture, we first boost the T.Rate source model trained with limited data (*e.g.* 5 or 7 users) with training techniques commonly used in current computer vision applications, *i.e.* data augmentation and model pretraining. Without loss of generality, in this paper, we leverage AutoAugment Cubuk et al. (2018) and ImageNet pretrain to boost the ResNet50. From (c) of Figure 2, we can clearly see, with these techniques we can successfully increase the T.Rate of 1% and 2% of clients from around 3% to 36% and 48% respectively. With 7% or 10% of clients' data, the proposed attack setting achieves a high T.Rate of more than 80% (10% higher than the transfer-based attack). That is to say, with simple training techniques, malicious clients can attack with more than 40% of success rate with one or clients and 80% with 7 to 10 clients.

**Training substitute model in federated manner.** To further testify the conjecture mentioned above, we train the substitute model with same number of clients' data in federated manner. Specifically, since we have the knowledge of each samples' clients, we partition the collected data from malicious clients as the federated model trains the source model in federated manner. We discuss the importance of such knowledge to transfer attack in Appendix D. Results can be seen in (c) of Figure 2. We can see that, with federated source model, the T.Rate can be slightly boosted at the beginning (limited number of clients) but continue to increase as the number of users increases and finally

contributes to a significantly high T.Rate of 99%. This demonstrates our conjecture regarding the trend and also shows that if the hostile party trains the substitute model in federated manner, the T.Rate can be further increased despite of its computation burden.

## 5 TWO INTRINSIC PROPERTIES CONTRIBUTING TO TRANSFER ROBUSTNESS

To fully understand how adversarial examples transfers between centralized and federated model, we study two intrinsic properties of FL and its relation with transfer robustness. To protect the privacy of clients and leverage the massive data from user-end, FL utilizes distributed data to train a global model through local updates and aggregation at the server McMahan et al. (2017). As consequence, the distributed data and the aggregation operation is the core component of a FL method.

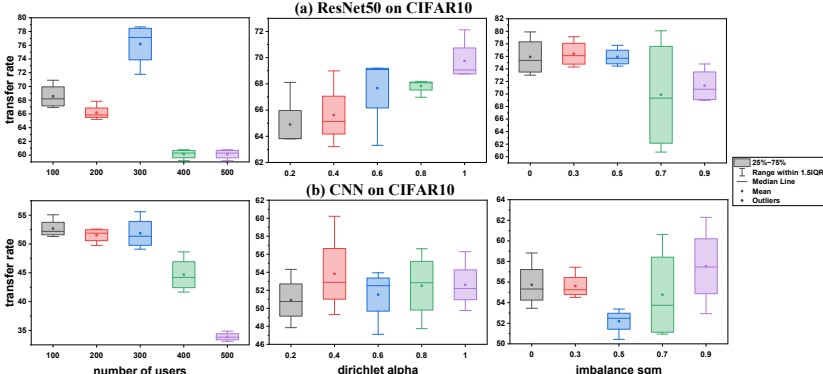

Figure 3: T.Rate vs. data of different heterogeneity and dispersion degree. (a): the top 3 are results of ResNet50; (b): the bottom 3 are results of CNN; Left: T.Rate as a function of number of users in federated training; Middle: T.Rate as a function of dirichlet alpha; Right:T.Rate as a function of unbalanced sgm.

### 5.1 DECENTRALIZED TRAINING AND DATA HETEROGENEITY

The research goal of this section is to explore the relation of decentralized data training and transfer robustness. That is, we want to see if the degree of dispersion and the heterogeneity of distributed data affects the transferability of adversarial examples generated to attack the federated model.

**Control the degree of dispersion and heterogeneity.** To explore the impact of distributed data on adversarial transferability, we control the decentralization and heterogeneity through four indexes. By varying the number of clients in the partition, we alters the degree of dispersion of distributed data. For heterogeneity, we changes the number of maximum classes per client McMahan et al. (2017), the alpha value of Dirichlet distributions Wang et al. (2020); Yurochkin et al. (2019) and the log-normal variance of the Log-Normal distribution used in unbalanced partition Zeng et al. (2021). We leverage FedLab framework to generate the different partitons Zeng et al. (2021). For the rest experiments, we leverage the centralized trained model as the source to perform the transfer attack.

**Degree of decentralization reduces transferability.** We generate partition with different number of clients and train federated models on these partitions. The results can be seen in the left of Figure 3. We can see, despite fluctuations, the T.Rate drops as the number of users in the partition increases.

**Data heterogeneity affects transfer attack.** To control the degree of non-iid distribution, we generate non-iid partitions according to Dirichlet distribution with different $\alpha$. Smaller $\alpha$ means a more non-iid partitions. We report the T.Rate in the middle of Figure 3. We report the results on pathological non-iid partitions (controlled by the number of maximum classes per client) in Appendix B. We can see that with ResNet50, T.Rate increases clearly as $\alpha$ increases (less non-iid). We also explore whether unbalanced data leads to decrease of transferability as displayed in right of Figure 3. We generate unbalanced data according to Log-Normal distribution and larger variance (sgm) denotes more unbalanced partitions. To validate the correlation, we provide statistical testing for correlation coefficient in Appendix C. With Spearman correlation coefficient, we report a significant negative correlation on ResNet50 between log-normal variance and T.Rate under significance level of 0.1 ($p$-value=0.05). We report significant correlation on all results with a level of 0.05 except the CNN

experiments with different dirichlet $\alpha$ and unbalance sgm (Appendix C). To better demonstrate the correlation, we also report visualization results on linear regression for all of the above results in Appendix E. To sum up, we demonstrate the decentralized training reduces the transferability of adversarial examples. This can be attributed to the enlarged discrepancy of decision boundary between source model and target model when the training data is more decentralized or heterogeneous. The heterogeneous distribution leads to a loss landscape significantly different from the ones of iid and centralized model and also results in a loss landscape with high variance Caldarola et al. (2022), which all contribute to lower the transfer possibility of adversarial examples Demontis et al. (2019).

## 5.2 AVERAGING LEADS TO ROBUSTNESS

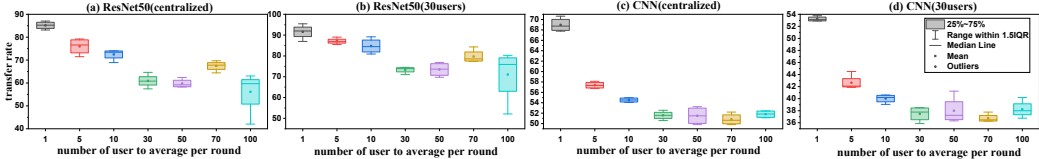

Figure 4: Transfer rate v.s. different number of clients selected to average in each round; (a) ResNet50 results with source model trained in centralized manner with full data; (b) ResNet50 results with source model trained with 30 users; (c) CNN results with source model trained in centralized manner with full data; (d) CNN results with source model trained trained with 30 users;

We explore the other core property, averaging operation, of FL and its correlation with transfer robustness. We use the FedAvg from McMahan et al. (2017) as base models. To change the degree of averaging in FL, we modify the number of clients selected to average at each round. We use centralized model and source model with 30 clients' data as substitute model to perform the transfer attack. Both models are early stopped at 90% as mentioned in Section 3.1. We plot the relation of T.Rate and the number of users in Figure 4. Both CNN and ResNet50 exhibits a decreasing trend as the number of clients used to average increases. We also provide a statistical testing to validate the correlation in Appendix C. With Spearman correlation coefficient, we report a significant negative correlation on all four experiments (all p-values are less than .001). This demonstrates that averaging in FedAvg improves the robustness of federated model against transfer attack.

## 5.3 SUMMARY

We list the results of our investigations as the below take-home messages and their implications:

- The heterogeneous data and large degree of decentralization both result in lower transferability of adversarial examples from the substitute model. → The attacker can benefit from closing the discrepancy of substitute model and target model (*e.g.* train substitute model also in federated manner) in terms of transferring.
- With more clients to average at each round, the federated model becomes increasingly robust to black-box attack. → Defenders can benefit from increasing the number of clients selected at each round to average.

In addition, we also identify a different, simpler, but practical attack evaluation for FL, which can serve as the standard robustness evaluation for future FL applications.

## 6 CONCLUSION

Federated Learning provides one of the most promising solutions to leverage the massive data emerging from user-end. Various topics regarding its robustness and security issue have been raised and discussed. In this paper, we identify a different, simpler, but practical setting: a group of malicious clients have participated in the training acting as benign clients, and only reveal their adversary position after the training to conduct transferable adversarial attacks with their data. We evaluate the possibility of such attack settings and find that with limited data we can achieve similar transfer rate as transfer attack using full training dataset. We also explore the correlation between two intrinsic properties and the transfer robustness of FL and discover that decentralized training, heterogeneous data and the averaging operation contributes to transfer robustness and significantly decrease the transferability of adversarial examples.

ETHICS STATEMENT

In this work, we aim to further understand the security challenges a Federated Learning system can face in reality with a new and practical settings. We believe our study will benefit the society with a full understanding of the working mechanism of the potential attacks in the scenario raised in the paper, and thus build the ground for continued study of the methods that are secure against such threats.

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
