# OpenReview forum: "Understanding Adversarial Transferability in Federated Learning"
_ICLR.cc/2023/Conference — Submitted to ICLR 2023_

### Official Review · Reviewer_bp6y · 2022-10-21

**Confidence:** 4
**Correctness:** 2
**Technical Novelty And Significance:** 2
**Empirical Novelty And Significance:** 1
**Recommendation:** 5

**Clarity, Quality, Novelty And Reproducibility:**

- Poorly motivated threat model, why would the attacker perform a transfer attack when FL protocol when there are malicious clients who observe the victim model at every FL round?

- Experiments are insufficient, vanilla PGD is not particularly transferable.

**Strength And Weaknesses:**

+ Covert attackers using FL training for reconnaissance and performing attacks later on is a totally valid threat model that is significant enough to study.


**Summary Of The Paper:**

This paper studies the transfer attacks in the context of federated learning (FL) where the attackers don't act maliciously during training but use the information they obtained to perform transfer-based adversarial attacks later on. The paper provides quantitative and qualitative results on the factors that affect the attack success on a simple FL setting.

**Summary Of The Review:**

1) The proposed threat model doesn't make too much sense to me. Malicious clients don't do anything but they observe the model at every round of FL to be able to send the server the models updates on their data. So, these clients have access to the actual victim model (or something very close to it from the later rounds of FL). Why would they then train another model to launch transfer attacks on the victim model? Can't they just performing an almoist white-box attack? Please tell me if I'm missing something, I'm willing to update my review.

If the setting is that the actual attacker knows the data of some of the clients (without being able to observe the FL models these clients train), then we can't call the clients malicious anymore, if anything they're compromised. In this case, I still can't say the contributions of this paper are significant enough because there's plenty of work studying how training set knowledge affects transferability [1,2]. Although these papers study centralized training, I don't think the results will be different enough to justify a new paper.

2) A more interesting setting would be that the malicious clients are not used throughout the FL training. They participate in the first N round and become inactive afterwards. So, the malicious clients don't have access to final victim model but an intermediate model from the Nth round. Can the attacker leverage this intermediate model + data from the clients to improve the trainsferability to the final victim model? How does the attack success change as a function of N?

3) I would recommend using attacks other than PGD. There's a lot of work on designing transferable attacks in various settings [3,4]. The risk of using PGD is that it probably underestimates the transfer success.


[1] https://www.usenix.org/conference/usenixsecurity18/presentation/suciu
[2] https://www.usenix.org/conference/usenixsecurity19/presentation/demontis
[3] https://openaccess.thecvf.com/content_CVPR_2019/papers/Inkawhich_Feature_Space_Perturbations_Yield_More_Transferable_Adversarial_Examples_CVPR_2019_paper.pdf
[4] https://openreview.net/forum?id=BJlRs34Fvr

---

> ### Author Response · Authors · 2022-11-15
> **Response to Reviewer bp6y**
>
> We sincerely thank the reviewer for the insightful and constructive comments.
>
> ```>>> Q1``` Unreasonable setting
>
> ```>>> A1``` We thank the reviewer for pointing out this concern, which is also raised by the Reviewer wDFS. From the scope of research, the participants of FL system do have access to the global model thus making the attacker easier to attack. However, as discussed in the Discussion of Section 2, we attribute the significance of this setting to two reasons.
> * The adversary might compromise a few clients obtaining their data to launch the attack, which poses an additional threat to FL compared with a centralized service. After acquiring the data originally used for training, the adversary can launch the attack more easily (compared with the black-box attack in a centralized setting) alone or combined with query-based attacks (as shown in Appendix A).
> * Despite, ideally, each participant having access to the global model, in real-world applications (e.g. Gboard [1]), the infrastructure provider will impose additional protection such as encryption or encapsulation over the local training. For instance, Gboard from Google provides the next-word prediction with FL, which requires users to install an app to participate. For an adversary, it's impossible to obtain the global model without breaking or hacking the app or hijacking and decrypting the communication, despite all the things happening "locally". We believe this is much more difficult and laborsome than our setting which significantly boosts the transferability by simply acting as a benign.
>
> ```>>> Q2``` More interesting setting
>
> ```>>> A2``` We sincerely thank the reviewer for raising this new interesting setting. Indeed, if the malicious client participates in the first few rounds and becomes inactive, the possibility of an attack with an outdated model is questionable and worth investigating. We will additionally conduct experiments to validate this attack setting in the future due to limited time.
>
> ```>>> Q3``` Better transfer attack other than PGD
>
> ```>>> A3``` We thank the reviewer for the sincere suggestions. Since the purpose is our study is to explore the possibility of a transfer attack, we leverage the basic threat model (widely adopted PGD) to perform the attack. We understand the concern raised by the reviewer and will improve the experiment further.

---

> > ### Comment · Reviewer_bp6y · 2022-11-24
> > **Thanks for clarifying.**
> >
> > 1I appreciate the clarifications about the threat model. Please update the manuscript to carefully reflect this. I'm going to boost my score to 5.
> >
> > One question, though: your threat model says that the FL clients are not malicious but their data is leaked. An attacker who can compromise the whole data set of a client, cannot be a weak attacker in practice so obtaining the global model by "breaking or hacking the app or hijacking and decrypting the communication" is not outside the realm of possibility. I cannot think of a realistic attacker who can do one but not the other. If you can come up with something convincing please let me know.
> >
> > My suggestion would be that you assume some of the data (e.g., 20%) of some clients are compromised but these compromised clients still have data that's kept secret from the attacker which is used for FL. For example, the attacker can use the photos they're sharing on social media which then are used in FL to train an image recognition model but not all photos of a client will be on social media.
> >
> > Ultimately, the intellectual contributions of this paper are reasonable and I think are interesting. But I'm not convinced that this threat model is interesting/realistic enough to justify a publication at ICLR.

---

### Official Review · Reviewer_wDFS · 2022-10-24

**Confidence:** 5
**Correctness:** 2
**Technical Novelty And Significance:** 1
**Empirical Novelty And Significance:** 2
**Recommendation:** 3

**Clarity, Quality, Novelty And Reproducibility:**

The paper is not novel.
The code is not given, thus I don’t think it is easy to reproduce.


**Strength And Weaknesses:**

Strength:
- The research topic is important. Defending adversarial attacks in FL is useful in reality.

Weakness：
- The setting is unreasonable. The paper assumes the attacker controls some clients that can take part in the FL training and considers black-box attack where the attacker cannot access the model (i.e., does not know the model parameters). Since the malicious clients can take part in the training, they definitely know the model parameters, which contradicts with black-box attack.

- Table 1 is confusing.
    - (1) The paper says “we early stop at 85% for adversarial training”, but the reported acc of R50 (adv) is 81.23 (centralized) and 80.05 (federated).
    - (2) Since same-acc shows the early-stopped results, it should be lower than regular. However, in federated CNN (adv), the adv. acc of same-acc (25.5) is higher than regular (24.35). Can you explain why?
    - (3) The paper conducts a “fair” comparison (same-acc) and says “when both models reach a comparable clean accuracy, federated model shows greater robustness against white-box attack compared with the centralized model.”I don’t think this experiment is fair: no evidence proves that when both models reach a comparable clean accuracy, they reach the same training stage. Previous studies [1] also show robustness may be at odds with accuracy. Thus, there is no strong relationship between acc and adv. acc.


- The paper conducts extensive experiments but lacks theoretical analysis.


[1] Tsipras D, Santurkar S, Engstrom L, et al. Robustness may be at odds with accuracy[J]. arXiv preprint arXiv:1805.12152, 2018.

**Summary Of The Paper:**

The paper investigates black-box adversarial attacks in FL. The paper assumes the attackers control some clients, thus have access to some training data. The attackers can launch transfer-based black-box attack with the training data. The paper has implications for understanding the robustness of federated learning systems and poses a practical question for federated learning applications.

**Summary Of The Review:**

The paper considers transfer-based black-box adversarial attack in FL. However, the setting is unrealistic. The experiments are confusing. Thus， I think the paper is below the bar of ICLR.

---

> ### Author Response · Authors · 2022-11-15
> **Response to Reviewer wDFS**
>
> We thank the reviewer for the constructive comments and thank the reviewer for considering the research topic is important and useful.
>
> `>>> Q1` Unreasonable setting
>
> `>>> A1` Thanks for pointing out this concern. From the scope of research, the participants of FL system do have access to the global model thus making the attacker easier to attack. However, as discussed in the Discussion of Section 2, we attribute the significance of this setting to two reasons.
> * The adversary might compromise a few clients obtaining their data to launch the attack, which poses an additional threat to FL compared with a centralized service. After acquiring the data originally used for training, the adversary can launch the attack more easily (compared with the black-box attack in a centralized setting) alone or combined with query-based attacks (as shown in Appendix A).
> * Despite, ideally, each participant having access to the global model, in real-world applications (e.g. Gboard [1]), the infrastructure provider will impose additional protection such as encryption or encapsulation over the local training. For instance, Gboard from Google provides the next-word prediction with FL, which requires users to install an app to participate. For an adversary, it's impossible to obtain the global model without breaking or hacking the app or hijacking and decrypting the communication, despite all the things happening "locally". We believe this is much more difficult and laborsome than our setting which significantly boosts the transferability by simply acting as a benign.
>
> `>>> Q2.1` Confusing early stop
> `>>> A2.1` Thanks for pointing out this problem. The early stop criteria for adversarial training of ResNet50 is 80\%. We will revise the paper to correct this as in the newly uploaded version.
>
> `>>> Q2.2` With federated CNN (adv),  the adv. acc of same-acc (25.5) is higher than regular (24.35).
>
> `>>> A2.2` We thank the reviewer for the question and concern. The reason for this is that we train the CNN with PGD for 400 rounds for the regular-acc while the same-acc model is also early stopped at around 400 rounds (when it reaches 72\% acc). We also find that the adversarial training converges at around 24\% ~ 25\% adv.acc, thus longer training does not lead to significantly better adv.acc, as shown in the second table. Thus, the problem that the same-acc is higher than regular is probably due to fluctuations brought by different random seeds. We additionally run experiments of same-acc (early stopped at 70\%) and re-run experiments of same-acc (early stopped at 72\%) 3 times and average the results as followed:
>
> |  Paradigm  | epochs / rounds |same-acc | same-adv.acc |
> |  ----  | ----  |  ----  |  ----  |
> | centralized (stopped at 72\%)  | 19 | 72.38 | 20.36 |
> | federated (stopped at 72\%) | 397 | 72.16 | 24.17 |
> | centralized (stopped at 70\%)  | 14 | 70.53 | 18.02 |
> | federated (stopped at 70\%) | 175 | 70.50 | 22.74 |
>
> |  Paradigm  | epochs / rounds | regular-acc | regular-adv.acc |
> |  ----  | ----  |  ----  |  ----  |
> | federated | 400 | 72.87 |  24.35 |
> | federated | 600 | 72.98 | 24.01  |
> | federated | 800 | 73.23 | 24.92  |
>
>
> `>>> Q2.3` No strong relationship between acc and adv. acc
>
> `>>> A2.3` We understand the concern raised by the reviewer. Robustness is influenced by the extent of convergence [2]. Despite there is no strong relationship between acc and adv. acc, we argue that the accuracy can partially reflect the training stage of the model, which is relatively better than, say, a pre-defined number of training epochs without any indicator as in [3], since we intend to compare the robustness of different training paradigms which requires different training time to converge. We also understand the problem in such a comparison and this is the reason why we also present the "same-acc" for reference. We are open to discussion of any criteria that conduct a fairer comparison.
>
> `>>> Q3` A lack of theoretical analysis
>
> `>>> A3` In this paper, we focus on the empirical evaluation of (i) the possibility of a transfer attack with limited data and (ii) how different degree of decentralization and the heterogeneity of data
> affects the transfer attack, thus the theoretical analysis is out of the scope of this work. However, as the empirical evaluation reveals an important security challenge as well as some intriguing findings regarding the transfer robustness of the FL model, we will further add the relevant theoretical analysis to the future revision.
>
> [1] Hard A, Rao K, Mathews R, et al. Federated learning for mobile keyboard prediction[J]. arXiv preprint arXiv:1811.03604, 2018.
> [2] Rice L, Wong E, Kolter Z. Overfitting in adversarially robust deep learning[C]//International Conference on Machine Learning. PMLR, 2020: 8093-8104.
> [3] Zizzo G, Rawat A, Sinn M, et al. Fat: Federated adversarial training[J]. arXiv preprint arXiv:2012.01791, 2020.

---

### Official Review · Reviewer_zdkS · 2022-10-27

**Confidence:** 3
**Correctness:** 3
**Technical Novelty And Significance:** 1
**Empirical Novelty And Significance:** 3
**Recommendation:** 5

**Clarity, Quality, Novelty And Reproducibility:**

The experiment details are clear enough for reproducibility.
The writing is in general good.

**Strength And Weaknesses:**

Strength: The paper discusses an interesting and important matter. It conducts detailed simulation studies, aiming to reveal how transferability is affected by FL learning.

Weakness: It seems that the results from CNN and ResNe50 are not consistent, hurting the credibility of the author's arguments. Such inconsistency somehow suggests that architecture plays a role.

Definition of transfer accuracy (T.Acc): what is ||x||? Although all simulation presents the results about the T.Acc, there is no discussion on the trend of T.Acc. Then what is the purpose of introducing this notation?

Please double check the legend in figure 2. The paper claims that "T.Rate against the federated model increases till 20 users and decreases.", but the figure shows that T.Rate against the centralized model increases till 20 users and decreases.

As an empirical work, I suppose the current scale of the experiment is not big enough to convince readers.

Figures in Section 5, i.e. Fig 3 & 4, display a complicated trend rough than the claimed monotone changing. It seems some in-depth messages are missing.

**Summary Of The Paper:**

This paper delivers an empirical assessment of adversarial transferability between FL trained model and centralized models, between limited data trained model and full data trained model. Based on the empirical evidence, it conjectures that FL models are more robust and their attack possesses high transferability, and believe it is caused by two factors: the decentralized training on distributed data and the averaging operation.

**Summary Of The Review:**

I appreciate this attempt to understand adversarial transferability in FL setting. However, I sense the current empirical evidence is not convincing enough.

---

> ### Author Response · Authors · 2022-11-15
> **Response to Reviewer zdkS**
>
> We thank the reviewer for acknowledging our problem. We also thank the reviewer for considering our paper clear to reproduce and well-written.
>
> ```>>> Q1``` The inconsistent results of CNN and ResNet50.
>
> ```>>> A1``` We would like to remind the reviewer that in the study of the transferability of adversarial examples, architecture does play an important role as pointed out by many previous studies [1, 2, 3]. Thus we consider it reasonable to observe different phenomena presented by different architectures. Despite the inconsistent behavior from CNN and ResNet in Figure 2, what we would like to emphasize is the possibility of attacking with limited data. As for experiments in Section 5, we show that despite some fluctuations, the general trends of CNN and ResNet are similar demonstrating that both decentralized or non-independent and identical distribution and averaging operation help improve the robustness of the federated model.
>
> ```>>> Q2``` Definition of transfer accuracy (T.Acc) and a lack of discussion.
>
> ```>>> A2``` We thank the reviewer for the question. The \|\| $\cdot$ \|\| notation denotes the cardinality of a set, which we define in the line "where \|\| $\cdot$ \|\| denotes the cardinality of one set" of section 3.2. As for the lack of discussion, the reason is that the trends of transfer accuracy are similar as the transfer rate but in an opposite way. We will additional revise our paper and add the discussion on transfer accuracy.
>
> ```>>> Q3``` Wrong Legend in Figure 2.
>
> ```>>> A3``` Thanks for pointing this out. We will revise it as in the newly uploaded version.
>
> ```>>> Q4``` Limited experiments.
>
> ```>>> A4``` We have additionally conducted experiments on CIFAR100 and FEMNIST to demonstrate our argument but do not display it due to limited space. We will add these experiments to revise our paper.
>
> ```>>> Q5``` Complicated trend in Section 5 and Figure 3, 4
>
> ```>>> A5``` In section5 of this paper, we mainly focus on the relation between the degree of non-IID or decentralization and the transfer robustness (i.e. whether the degree of dispersion and the heterogeneity affects the transferability of adversarial examples generated to attack the federated model). Thus, we believe the monotone relation (positive/negative correlation) is sufficient to illustrate such relations despite that there may be a more complicated trend, which is out of the scope of this paper.
>
>
> [1] Waseda F, Nishikawa S, Le T N, et al. Closer Look at the Transferability of Adversarial Examples: How They Fool Different Models Differently[J]. arXiv preprint arXiv:2112.14337, 2021.
> [2] Liu Y, Chen X, Liu C, et al. Delving into transferable adversarial examples and black-box attacks[J]. arXiv preprint arXiv:1611.02770, 2016.
> [3] Wu L, Zhu Z. Towards understanding and improving the transferability of adversarial examples in deep neural networks[C]//Asian Conference on Machine Learning. PMLR, 2020: 837-850.

---

### Author Response · Authors · 2022-11-15
**Manuscript Revision**

Dear Reviewers,

We just uploaded the revised manuscript according to the reviews, including the legend problem and the early stop criteria problem. Thanks again for your effort in reviewing this paper. If you have any further comments or concerns, please feel free to bring them up.

Best Regards.

---

### Decision · Program_Chairs · 2023-01-20

**Decision:**

Reject

**Justification For Why Not Higher Score:**

The presented empirical evidences are not convincing enough and the conclusions may be misleading.

**Justification For Why Not Lower Score:**

N/A

**Metareview: Summary, Strengths And Weaknesses:**

The paper delivers an empirical study to understand the transferability of adversarial robustness under federated learning. The reviewers found that the presented empirical evidences are not convincing enough and the conclusions may be misleading. For example, "the current scale of the experiment is not big enough to convince readers" and "the attacker controls some clients that can take part in the FL training and considers black-box attack where the attacker cannot access the model" (this concern is actually shared by two reviewers wDFS and bp6y). Thus, we cannot accept it for publication. If the authors would like to revise the paper and try another machine learning conference, I suggest the authors to carefully consider the problem setting to make it more realistic in the revised manuscript.